# A Novel Head Mounted Display Based Methodology for Balance Evaluation and Rehabilitation

**Eun-Young Lee, Van Thanh Tran and Dongho Kim ***

Graduate School, Soongsil University, 369 Sangdo-ro, Dongjak-gu, Seoul 06978, Korea; ella@gsclab.kr (E.-Y.L.); thanhit08@magiclab.kr (V.T.T.)
* Correspondence: cg@su.ac.kr; Tel.: +82-2-821-2889

**Abstract:** In this paper, we present a new augmented reality (AR) head mounted display (HMD)-based balance rehabilitation method. This method assesses the individual's postural stability quantitatively by measuring head movement via the inertial measurement unit sensor integrated in the AR HMD. In addition, it provides visual feedback to train through holographic objects, which interacts with the head position in real-time. We implemented applications for Microsoft HoloLens and conducted experiments with eight participants to verify the method we proposed. Participants performed each of three postural tasks three times depending on the presence or absence of augmented reality, the center of pressure (COP) displacement was measured through the Wii Balance Board, and the head displacement was measured through the HoloLens. There are significant correlations ($p < 0.05$) between COP and head displacement and significant differences ($p < 0.05$) between with/without AR feedback, although most of them were not statistically significant likely due to the small sample. Despite the results, we confirmed the applicability and potential of the AR HMD-based balance rehabilitation method we proposed. We expect the proposed method could be used as a convenient and effective rehabilitation tool for both patients and therapists in the future.

**Keywords:** augmented reality; head mounted display; postural stability; balance rehabilitation; balance assessment; visual feedback

## 1. Introduction

Falls and fall-related injuries are more fatal to people aged over 65 [1]. Fall-related injuries include long-term effects, high incidence, and significant costs [2], and the costs of non-fatal and fatal falls combined are estimated at approximately $50 billion in the US [3]. Sequelae of falls include pain, fear of falling, and loss of confidence [4,5]. These physical and psychological problems lead to morbidity, declined functional capacity, premature institutionalization, limitation in social participation, and disability [4–6]. Elderly who sustained fall-related injuries or have fallen obtain a poor quality of life in most cases [7,8].

An individual can achieve postural stability only if there are integration and coordination in harmony between central and peripheral factors such as vision, somatosensation, vestibular sensation, motor output, and musculature [9–12]. The central nervous system integrates information from various organs and formulates appropriate responses. The signals from these systems go to the brain and then serve reflexes to provide postural stability [13]. Afterward, it will direct and coordinate the musculoskeletal system to perform the appropriate movements to maintain balance [14]. Balance dysfunction includes a reduction of movement precision and environmental perception owing to impairment in vision, somatosensory, and vestibular information. Moreover, these defects affect balance and cause falls. Balance dysfunctions are general reasons for falls in the elderly [15].

Computerized dynamic posturography (CDP) was developed to provide a consistent quantitative postural control assessment, which is a computerized device used to quantify an individual's postural control through the movement of the body's center of gravity using dynamometric platforms [16,17]. CDP detects postural sway by measuring shifts in the center of pressure (COP) as a user moves within their limits of stability. It is an integral component in the diagnostic assessment of balance impairments to help to identify the underlying sensory and motor control impairments [17,18]. Postural strategies can be quantified to static and dynamic perturbations by determining whether the user uses postural strategies during the test in response to postural disturbances. Clinicians use the data from CDP to evaluate the contribution of neuromuscular systems and sensory information to postural stability [19].

Using biofeedback training has shown to be a promising method to deliver balance rehabilitation [20]. Previous meta-analyses have reported that multifactorial intervention involving postural control training could reduce falls [21,22]. With single intervention, some traditional balance exercises reduced the risk of falls in community-dwelling older adults [23]. Augmented Reality (AR) is a type of virtual reality that provides a view of the tangible world augmented by, and/or spatially registered with computer-generated information [24]. Augmented feedback is frequently used to aid training of motor skills by providing patients with information about the skill not appreciated by their own sensory feedback [25–27]. Representative visual feedback-based balance training is to have patients stand on a force plate while providing continuous visual feedback of the COP and directing patients to minimize movement of the COP [28–30], and the intervention is ordinary in clinical practice [31]. Balance-impaired people tend to rely on vision and exteroceptive information on postural control [32]. In this respect, visual feedback-based balance training can enhance sensorimotor integration by recalibrating of the sensory inputs contributing to postural control [33]. This training shows to be effective for postural sway reduction and balance control improvement among the elderly [34,35].

Recently, virtual reality and active video games promise rehabilitation tools because of their potential to facilitate motivating, ample, and feedback-rich practice [36,37]. Developments in the gaming industry have ensured integration of users' physical movements in virtual environments, providing new opportunities to offer precise augmented visual feedback that engages the user in the motor and cognitive tasks simultaneously [38]. Commercial game consoles with games that target postural control and other forms of functional tasks (e.g., Nintendo Wii™Fit) are increasingly used in rehabilitation [39]. A steady increase in the number of peer-reviewed articles evaluating the effects of these interventions in many rehabilitation populations has been observed over the past 20 years. This increase reflects a growing interest in virtual reality and serious games from the rehabilitation research and development sectors. These systems were used for various patients, practice settings, and functional recovery goals by clinicians and therapists [40].

Following this trend, various attempts are being made to apply immersive virtual reality and immersive augmented reality to rehabilitation. Especially, Head Mounted Displays (HMD) offer an efficient and cost-effective alternative to other tools used to generate virtual environments for research, training, and treatment [41,42]. Some researchers developed the balance rehabilitation configured with HMDs with customized software and tracking hardware such as the Microsoft Kinect [43–45].

A user who wears a virtual reality HMD in front of the eyes cannot interact with objects and becomes difficult in a virtual environment [46]. For patients with balance dysfunction, users are visually disconnected from the real world, thus could feel extremely anxious in a fully immersive virtual environment with HMD. Therefore, training using augmented reality can represent a better alternative than training using virtual reality for patients with a balance disorder.

We will verify AR HMD as a balance assessment tool by analyzing the correlation with the results measured through the Wii Balance Board (WBB), which is validated for balance assessment [47]. Ultimately, the effectiveness of the training is going to be verified by comparing the head movement on the presence or absence of augmented reality.

## 2. Proposed Method

The proposed method is to provide a portable and easy-to-use yet reliable to support both therapists and patients for managing rehabilitation using augmented reality HMD (Figure 1). The balance rehabilitation procedure requires carrying out postural tasks in various conditions under the accurate observation of therapists. The first phase in the design process of our research is to understand the requirements of both balance-impaired patients and therapists. We discussed both therapists' needs and patients' difficulties, assuming the scenarios of evaluation and/or training of postural stability. In addition, we defined conditions for the postural stability assessment consisting of three stances, including double-leg stance, non-dominant leg stance, and tandem stance, and a reference posture.

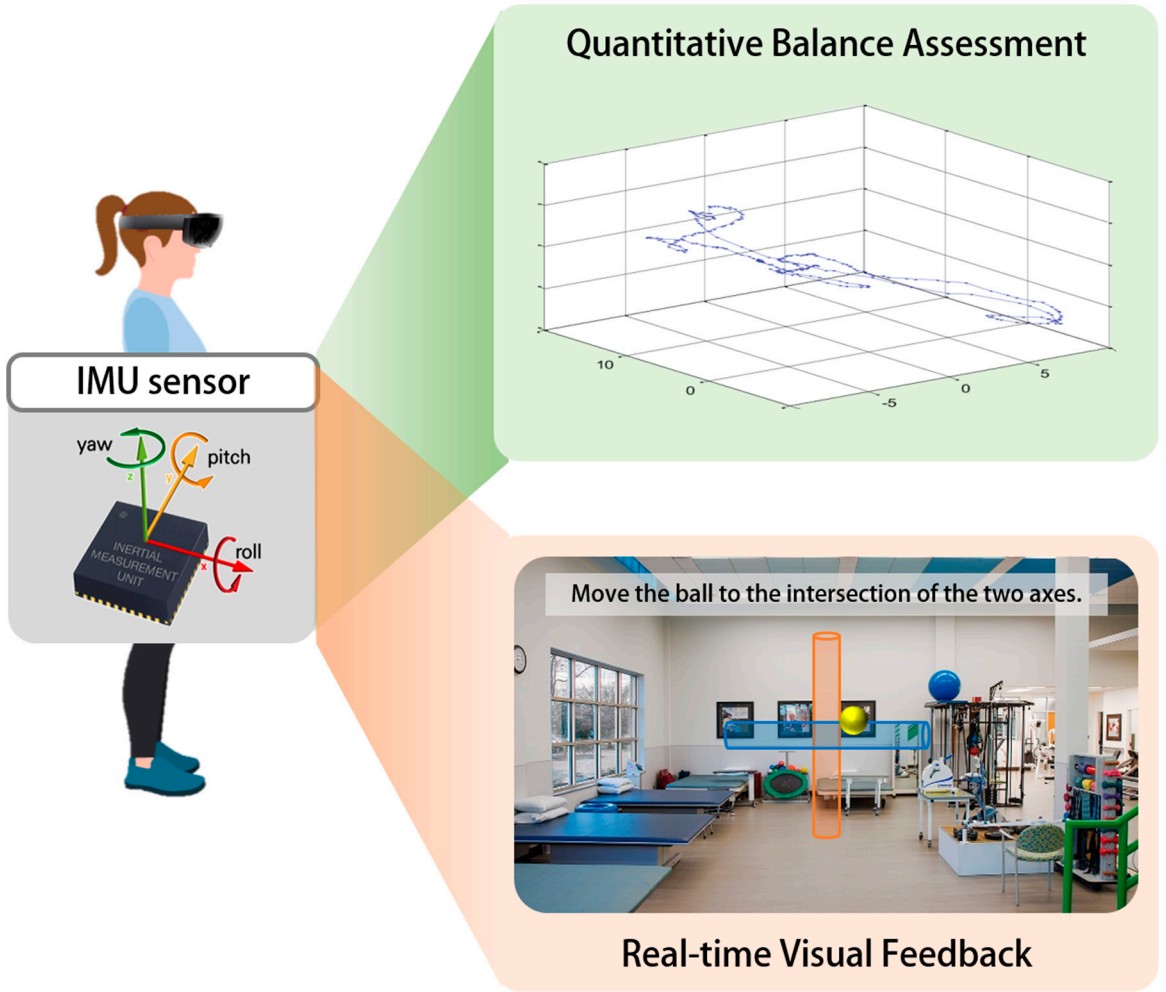

**Figure 1.** Proposed method.

Additionally, we assumed that postural instability could be estimated through head movements according to prior study [48–51]. In AR HMD, such as the Microsoft HoloLens (a stationary AR device), the head displacement can be easily sensed by the integrated inertial measurement unit (IMU) sensor. Each of the gyroscope, accelerometer, and magnetometer provides three-axis measurements.

Lastly, in order to provide appropriate feedback, we suggest augmented objects that respond to the user's movements in real-time. The therapist should encourage the patient to move the augmented object to the reference position during training while maintaining the reference posture. Repeating this process could reduce body sway in various postural tasks.

*Microsoft HoloLens for Balance Rehabilitation*

We used Microsoft HoloLens as it allows therapists and patients to satisfy their needs. HoloLens is a wearable augmented reality device that allows users to interact with holographic content augmented in the real world. To make augmented objects in the real world in a very realistic way, the holographic projection of the HoloLens display generates colored, multi-dimensional holograms. In addition, the HoloLens comprises cameras to capture information about the user's head orientation and position. HoloLens interprets gaze, gestures, and voice, enabling interaction with augmented objects, with spatial sound-synthesis. Especially, HoloLens contains an IMU that is validated for clinical practice [52], and a 2.4-megapixel photographic video camera, an energy-efficient depth camera, an ambient light sensor, and a four-microphone array. The IMU in HoloLens includes a gyroscope, an accelerometer, and a magnetometer, and is coupled with head tracking cameras, which enables HoloLens to understand where the user's head is and how it is moving.

HoloLens encourages users to sustain environmental awareness and combines the advantages of virtual reality with the ones allowed by the interaction with the real world. Furthermore, the patient can explore the physical space without obstacles or further limits in the room, which adds realism and engagement to the experience.

In the environment with HoloLens, patients are not isolated from the real world. It enables the patient to feel more comfortable and to interact with the therapist without difficulty. Moreover, HoloLens applications help therapists in integrating data about their intervention automatically.

HoloLens uses the orientation and position of the head to find out the gaze vector. Our method uses gaze cursor to recognize what the patient is targeting in the mixed reality world and to find out the patient's intent. To select an object, the user must perform the air-tap gesture by holding the thumb and index finger apart and then bringing them together. Voice recognition is another option for interaction, but it has not been registered in the current version because it is not convenient to non-English speakers.

We observed the recommendations for HoloLens applications development for positioning both the field of view of the camera (17 degrees) and message objects, such as the appropriate distances (not less than 0.8 m).

## 3. Materials and Experiment

### 3.1. Experimental Setup

In this study, we used the Wii Balance Board (Nintendo Co., Ltd., Kyoto, Japan) and HoloLens (Microsoft Corporation, WA, USA) to collect postural sway. Participants stood barefoot on the Wii Balance Board, wearing HoloLens during each measurement. The displacement of COP from WBB and displacement of the head from HoloLens were collected. Figure 2 illustrates the experimental setup.

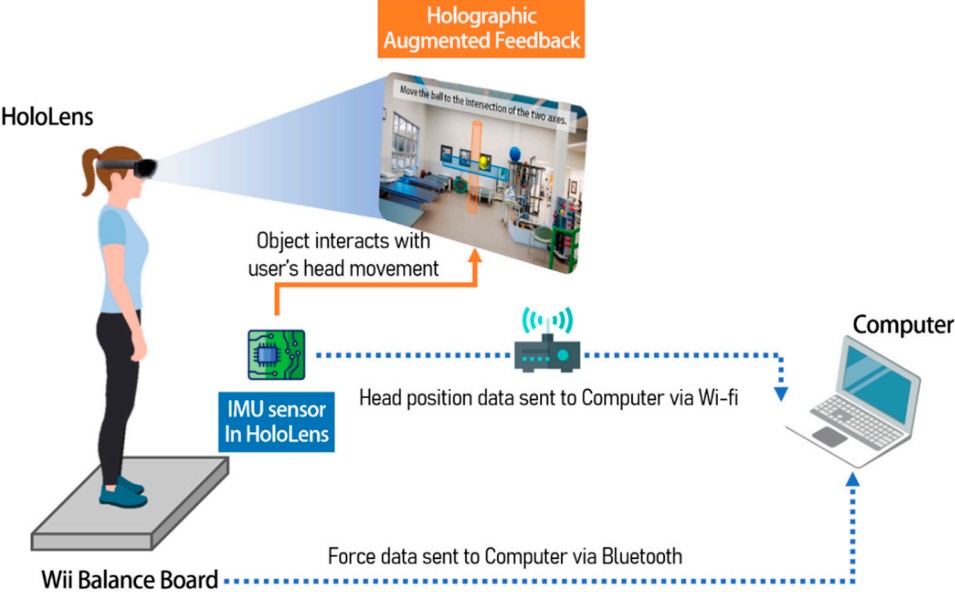

**Figure 2.** Experimental design.

### 3.2. Software Algorithms

The balance evaluation module consists of two applications, which communicate via a Bluetooth signal and Wi-Fi signal with a computer. The first application is developed for WBB based on a library called WiimoteLib [53]. This application runs on a computer and records values from sensors that measure the force of interaction from the user's foot. HoloLens runs the second application to collect data and to display augmented content. This application records the movement information of the user's head based on the device's sensors. Both applications are developed in C# programing language with Visual Studio 2017-integrated development environment from Microsoft. The three-dimensional movement from HoloLens and two-dimensional information from WBB will be collected during the preset test time. All data are obtained in real-time at the rate of 30 frames per second. The data are low pass filtered with a cut-off frequency of 15 Hz for noise reduction.

Before measurement, the coordinate of HoloLens and WBB must be calibrated. In the experiment with AR, a virtual object will be rendered in from of the user's eyes. Therefore, the user must stand the same position before and after starting the application to make sure that the calculation of the virtual object movement is correct. Pressing the "Start" button of the WBB application and pressing the button on the clicker—known as a remote-control device of HoloLens—will launch both applications at the same time.

#### 3.2.1. Center of Pressure Measurement

The WBB measures postural sway in the medial-lateral ($COP_x$) and anterior-posterior ($COP_y$) directions based on downward force sensor data according to the following formulas:

$$COP_x = 21.7 \times \frac{(F_{TR} + F_{BR}) - (F_{TL} + F_{BL})}{F_{TR} + F_{BR} + F_{TL} + F_{BL}}, \quad COP_y = 14.3 \frac{(F_{TR} + F_{TL}) - (F_{BR} + F_{BL})}{F_{TR} + F_{BR} + F_{TL} + F_{BL}}, \tag{1}$$

where $F_{TR}$, $F_{TL}$, $F_{BR}$, and $F_{BL}$ are the force sensor values from the top right, top left, bottom right, and bottom left corners of the WBB.

The total path distance of *COP* is the main measure describing the postural sway during balance assessment. This variable was calculated according to the following formula:

$$Total\ displacement\ of\ COP(cm)\ = \sum_{k=2}^{n} \sqrt{(COPx_k - COPx_{k-1})^2 + (COPy_k - COPy_{k-1})^2} \qquad (2)$$

where $COPx_k$ and $COPx_{k-1}$ are adjacent time points in the *COPx* time series, and $COPy_k$ and $COPy_{k-1}$ are adjacent time points in the *COPy* time series.

### 3.2.2. Head Displacement Measurement

The Hololens measures postural sway in the medial-lateral (H*x*), anterior-posterior (H*y*), and superior-inferior (H*z*) directions. The virtual game camera transformation is updated based on information from gyroscope sensors of HoloLens, including translation and rotation, when the user's head is moving. The position of the head is taken from the position of the virtual game camera in the application running on the HoloLens device as the following formula:

$$Camera_{cur} = Camera.main.transform.position$$
$$H_c = new\ Vector3(Camera_{cur}.x,\ Camera_{cur}.y,\ Camera_{cur}.z) \qquad (3)$$

where $Camera_{cur}$ is the current position of the virtual game camera, and $H_c$ is the current position of the head.

The sum of the user's head movement is calculated based on the position of the head in the previous frame and the current frame

$$Total\ displacement\ of\ head(cm)\ = \sum_{i=0}^{c} \sqrt[2]{\left(H_x^i - H_x^{i-1}\right)^2 + \left(H_y^i - H_y^{i-1}\right)^2 + \left(H_z^i - H_z^{i-1}\right)^2} \qquad (4)$$

where $H_x^i$, $H_y^i$, $H_z^i$ are the coordinate of the user's head at frame *x*.

### 3.2.3. Augmented Feedback

For augmented feedback, the process for measuring head movement is the same as the measurement, but there is a difference in that it provides visual feedback in real time. Figure 3 shows an example screenshot of HoloLens and it's description.

The application displays two virtual objects rendered in front of the user's eyes. The first object consists of two cylinders arranged perpendicularly representing the *x*-axis and *z*-axis of HoloLens coordinate system. The second object is a spherical object placed in the origin of the coordinate. The spherical object moves depending on the movement of the user's head as in the following formula:

$$O_c = O_p + \left(H_c - H_p\right) \qquad (5)$$

where $O_c$, $H_c$, $O_p$, $H_p$ are the current position and previous position of the spherical object and the user's head.

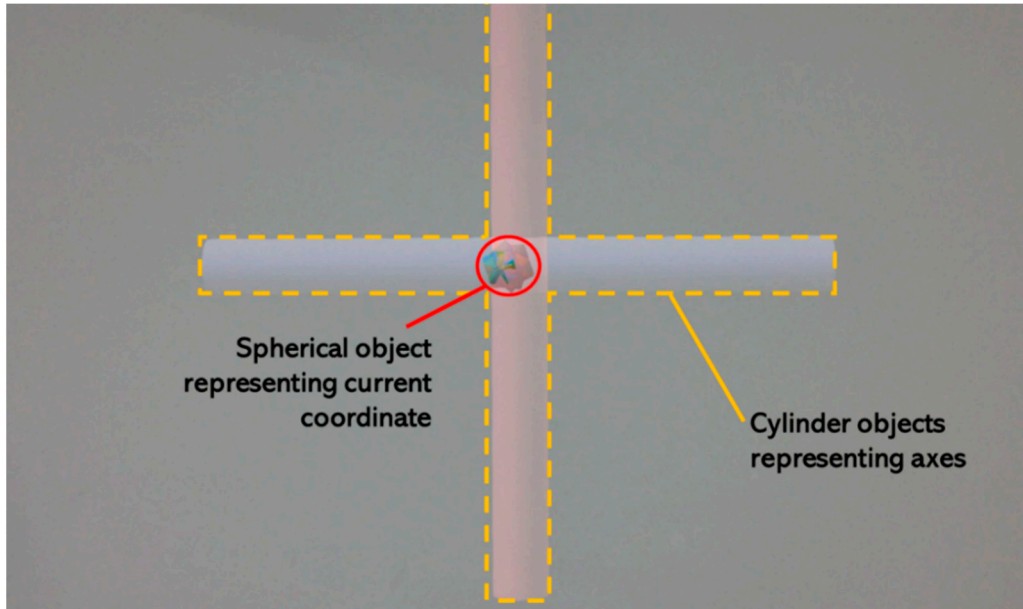

**Figure 3.** Screenshot from HoloLens in feedback application. The blue cylinder represents the *x*-axis of the HoloLens coordinate system, and the red cylinder represents the *z*-axis of the HoloLens coordinate system.

### 3.3. Ethical Statement

All experimental protocols were approved by the institutional review board at Soongsil University (SSU-201909-HR-143-01). Informed written consent was obtained from all participants prior to enrollment in this study.

### 3.4. Participants

Eight healthy male adults (mean ± SD age 24.75 ± 1.91 years, weight 70.13 ± 10.44 kg, and height of 171.38 ± 5.42 cm) were recruited from a university population for this study. We asked participants which leg they preferred for kicking a ball so that the knowledge of the participant's dominant leg was required for this study. All the participants were right footed (i.e., preferred to kick with the right foot). They reported no illnesses, injuries, or any neurological or musculoskeletal conditions that would affect their balance, and each participant received written informed consent about all procedures. The characteristics of the participants are summarized in Table 1.

**Table 1.** Participants characteristics.

|  | Mean | Standard Deviation |
|---|---|---|
| Age (years) | 24.75 | 1.91 |
| Height (cm) | 171.38 | 5.42 |
| Weight (kg) | 70.13 | 10.44 |
| BMI (kg/m$^2$) | 23.81 | 2.76 |

### 3.5. Procedure

One of the frequently used clinical tests for post-concussion assessment is the Balance Error Scoring System (BESS). The BESS measures postural instability by the examiner in the maintenance of various stances (feet together, single-leg, and tandem stance) by the patient, who is standing on different surfaces (firm and soft) and has his/her eyes closed. For this research, we used the modified BESS test, which includes only 3 stances on a firm surface (omitting stances on a soft surface).

Participants were instructed to wear the HoloLens and to stand barefoot on the WBB and keep their hands on their hips (iliac crest) in each test. We gave a familiarization period to allow participants to experience the three stances (Figure 4) and visual feedback conditions of the test.

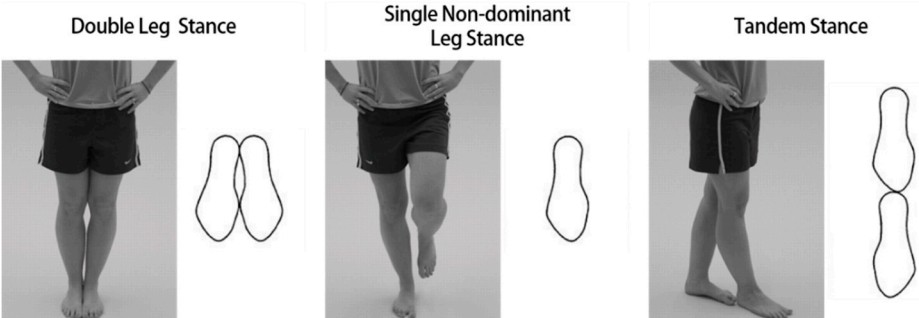

**Figure 4.** Three stances for the experiment.

Balance Tasks

- Double-leg stance without augmented feedback.
- Single-nondominant leg stance without augmented feedback.
- Tandem stance without augmented feedback.
- Double-leg stance with augmented feedback.
- Single-nondominant leg stance with augmented feedback.
- Tandem stance with augmented feedback.

Double-leg stance required that the participant keep both feet on the ground with their feet together. Single-leg stances were performed on the nondominant leg while lifting the foot of the other leg off the ground to an angle of approximately 45-degree knee flexion. During the tandem stance, the dominant foot is in front of the nondominant foot.

Following the preparations, participants performed each test condition three times in a randomized order, obtained from a computer-based random number generator. Once familiar with every condition, balance testing began. The participants heard a beep sound from the PC at the start and end of each test. Each trial lasted 20 s, and it began when the test position was held stable for several seconds. Between each test, participants were asked to sit down and take enough rest.

*3.6. Statistical Analysis*

The statistical analysis was carried out using the Python version 3.7 programming language and Pandas(https://pandas.pydata.org) and SciPy(http://scipy.org) package. Normality of the data was investigated using the Shapiro–Wilks test.

3.6.1. Correlation between COP and Head Displacement

We verified the HoloLens as a posture stability assessment tool by analyzing the correlation between the measurement results using the WBB and the HoloLens. We calculated the Pearson correlation coefficients to analyze the relationship between COP movement and head displacement in each test condition.

3.6.2. Effectiveness of Augmented Feedback

To detect effectiveness of augmented visual feedback, a paired sample t-test (parametric data) or Wilcoxon signed-rank test (parametric data) was used to quantify differences between the COP with and without AR feedback and differences between the head displacement with and without AR feedback in each stance.

## 4. Results

### 4.1. Correlation between COP and Head Displacement

In this study, the total path length of head movement in single leg stance without augmented visual feedback was positively associated with the COP (r = 0.481, $p < 0.05$). In addition, the Medio-Lateral displacement (r = 0.462, p < 0.05) and total path length (r = 0.426, $p < 0.05$) in the double leg stance with augmented visual feedback were also positively associated with the COP. Table 2 shows the correlations between head movement and COP in each test condition.

**Table 2.** Pearson correlation coefficients($r^2$) for the center of pressure (COP) and head movement in each test condition.

|  | $r^2$ |  | p |
|---|---|---|---|
| **Double leg stance without AR** |  |  |  |
| Medio-Lateral displacement(cm) | 0.218 |  | 0.306 |
| Anterior-Posterior displacement(cm) | 0.221 |  | 0.299 |
| Total path length(cm) | 0.071 |  | 0.742 |
| **Single leg stance without AR** |  |  |  |
| Medio-Lateral displacement(cm) | 0.075 |  | 0.729 |
| Anterior-Posterior displacement(cm) | 0.367 |  | 0.078 |
| Total path length(cm) | 0.481 | * | 0.017 |
| **Tandem stance without AR** |  |  |  |
| Medio-Lateral displacement(cm) | 0.234 |  | 0.271 |
| Anterior-Posterior displacement(cm) | −0.170 |  | 0.428 |
| Total path length(cm) | 0.189 |  | 0.377 |
| **Double leg stance with AR** |  |  |  |
| Medio-Lateral displacement(cm) | 0.462 | * | 0.023 |
| Anterior-Posterior displacement(cm) | 0.305 |  | 0.148 |
| Total path length(cm) | 0.426 | * | 0.038 |
| **Single leg stance with AR** |  |  |  |
| Medio-Lateral displacement(cm) | 0.002 |  | 0.994 |
| Anterior-Posterior displacement(cm) | 0.277 |  | 0.190 |
| Total path length(cm) | −0.016 |  | 0.940 |
| **Tandem stance with AR** |  |  |  |
| Medio-Lateral displacement(cm) | −0.145 |  | 0.499 |
| Anterior-Posterior displacement(cm) | 0.064 |  | 0.768 |
| Total path length(cm) | 0.182 |  | 0.394 |

Note: * $p < 0.05$.

### 4.2. Effectiveness of Augmented Feedback

All COP displacements with augmented visual feedback in all conditions were not significant when comparing with and visual feedback ($p \geq 0.05$) (Table 3). The results of the COP for each test condition are illustrated in Figure 5.

**Table 3.** Comparison of COP from the Wii Balance Board (WBB) between without/with Augmented Reality (AR) feedback.

|  | Without AR ± SD | With AR ± SD | *p*-Value |
|---|---|---|---|
| **Double leg stance** | | | |
| Medio-Lateral displacement (cm) | 23.24 ± 3.24 | 25.73 ± 6.09 | 0.129 |
| Anterior-Posterior displacement (cm) | 20.11 ± 5.01 | 19.43 ± 3.22 | 0.713 |
| Total path length (cm) | 34.24 ± 5.57 | 35.67 ± 7.02 | 0.485 |
| **Single leg stance** | | | |
| Medio-Lateral displacement (cm) | 68.37 ± 7.18 | 62.05 ±6.99 | 0.160 |
| Anterior-Posterior displacement (cm) | 63.60 ± 11.33 | 56.48 ± 4.57 | 0.098 |
| Total path length (cm) | 102.66 ± 13.40 | 92.19 ± 7.74 | 0.111 |
| **Tandem stance** | | | |
| Medio-Lateral displacement (cm) | 40.62 ± 5.68 | 43.13 ± 5.27 | 0.487 |
| Anterior-Posterior displacement (cm) | 33.59 ± 6.25 | 41.76 ± 10.67 | 0.100 |
| Total path length (cm) | 58.56 ± 8.43 | 66.70 ± 11.38 | 0.209 |

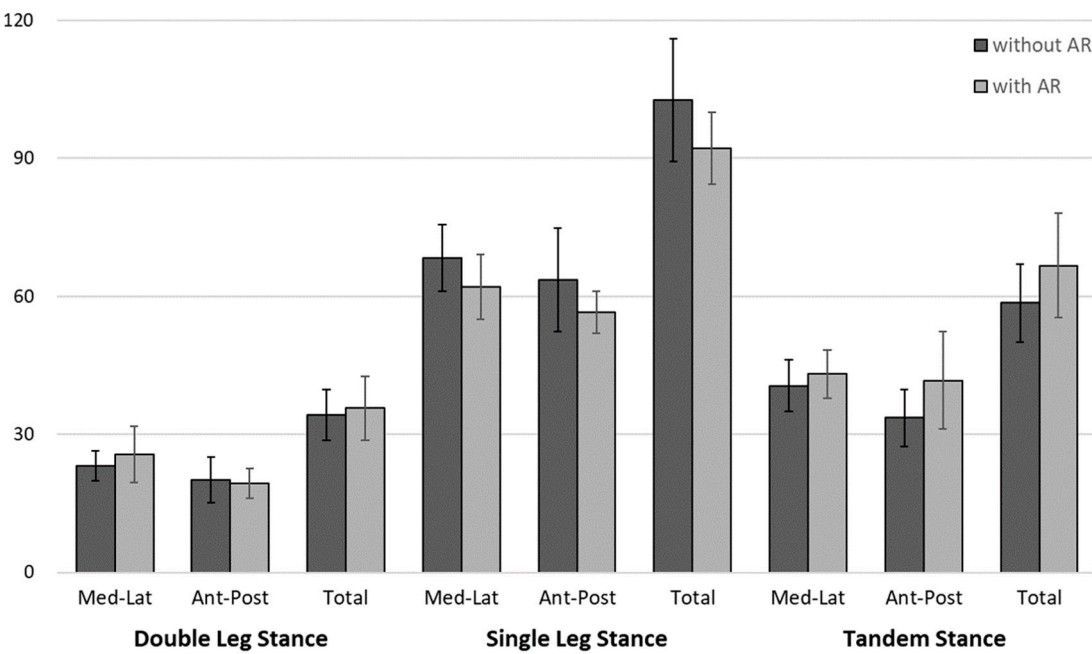

**Figure 5.** Comparison of COP displacement (cm) for each test condition.

Anterior-posterior head displacement in the single leg stance was significantly smaller at *p* < 0.05 than without feedback. (Table 4) All other head displacements were not significant when comparing without and with augmented visual feedback. The results of the head displacement for each test condition are illustrated in Figure 6.

**Table 4.** Comparison of the head movement from HoloLens between without/with AR feedback.

|  | Without AR ± SD | With AR ± SD | *p*-Value |
|---|---|---|---|
| **Double leg stance** | | | |
| Medio-Lateral displacement (cm) | 113.44 ± 22.59 | 121.69 ± 20.36 | 0.269 |
| Anterior-Posterior displacement (cm) | 49.12 ± 8.92 | 51.69 ± 12.68 | 1.000 |
| Total path length (cm) | 133.38 ± 22.00 | 142.50 ± 25.11 | 0.164 |
| **Single leg stance** | | | |
| Medio-Lateral displacement (cm) | 271.80 ± 98.52 | 234.92 ± 47.21 | 0.093 |
| Anterior-Posterior displacement (cm) | 134.58 ± 23.04 | 112.52 ± 18.24 | 0.020 * |
| Total path length (cm) | 330.73 ± 91.72 | 282.21 ± 48.64 | 0.056 |
| **Tandem stance** | | | |
| Medio-Lateral displacement (cm) | 175.01 ± 30.21 | 170.37 ± 26.48 | 0.758 |
| Anterior-Posterior displacement (cm) | 84.34 ± 15.54 | 82.17 ± 23.23 | 0.841 |
| Total path length (cm) | 211.00 ± 33.77 | 204.82 ± 31.26 | 0.484 |

Note: * *p* < 0.05.

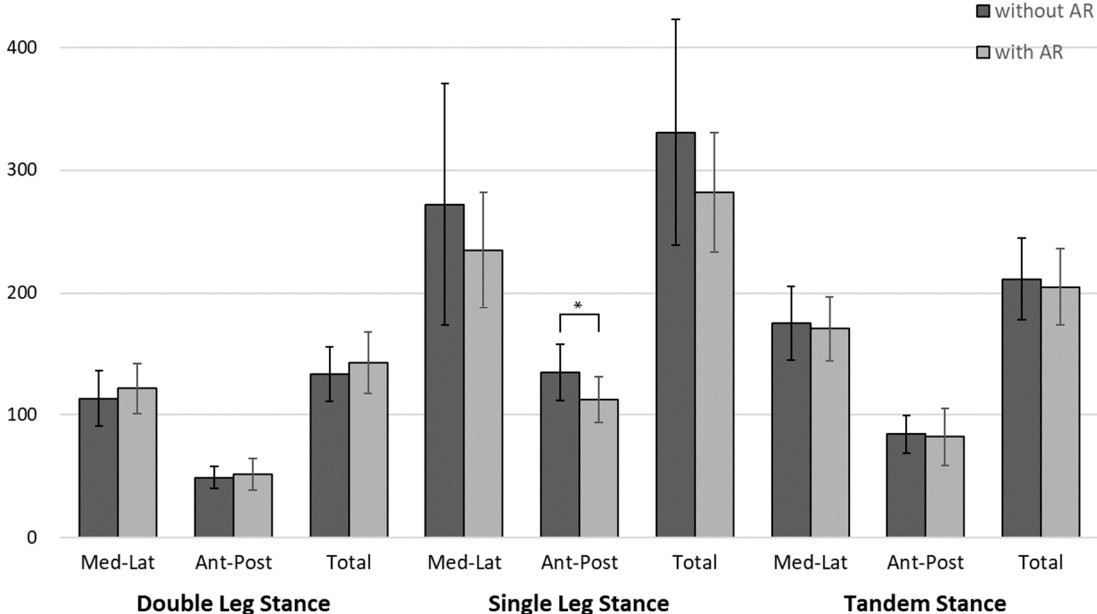

**Figure 6.** Comparison of head displacement (cm) for each test condition.

## 5. Discussion

In this paper, we suggested and implemented an AR HMD-based balance rehabilitation method. This method assesses the individual's postural stability quantitively by measuring the user's head movement via an IMU sensor integrated in AR HMD. In addition, it provides visual feedback to train through an augmented holographic object, which interacts with the user's head position in real-time. We also implemented applications for Microsoft HoloLens and conducted experiments with eight participants to verify the methods we proposed. Participants performed three test positions repeatedly depending on whether the augmented reality existed; the COP displacement was measured through the WBB, and the head displacement was measured through the HoloLens.

The head displacement is a measure used to assess balance performance and balance strategy. Terry et al. (2011) measured participants' head displacement and identified postural strategies using combined kinetic and kinematic data from the head [48]. Horak, Earhart, and Dietz (2001) tested the interaction of responses to combinations of support surface perturbations and the head. They reported body displacements were consistently in the opposite direction of the head. They noted

complex interactions of the somatosensory-evoked response to body displacement relative to the feet and the vestibulospinal-evoked response to head displacement [49]. De Nunzio, Nardone, and Schieppati (2005) analyzed the effect of a proprioceptive disturbance on the postural strategy through coordination between the displacements of the feet and head and temporal coupling between feet and head displacements [50]. Based on these studies, we assumed that the head movement and posture stability were correlated in our study. The correlation between COP measured by WBB and head displacement measured by AR HMD were analyzed to verify head displacement as an indicator of postural stability evaluation.

In this study, head displacement was moderate (between 0.4 and 0.6), positively associated COP with significance ($p < 0.05$) in the total path length in single leg stance without AR feedback, medial-lateral displacement in the double leg stance with augmented visual feedback and total path length in double leg stance with augmented visual feedback. Significant negative correlations were absent for all tests performed. It is noteworthy that there are two significant correlations (medio-lateral displacement and total path length) under the double leg stance with the AR condition only. One possible explanation behind this is because the double leg stance with AR is the most stable condition and this made similar measurement values during the test on both HoloLens and Balance Board.

Various approaches have been made for the intervention using visual feedback for balance, and these approaches are popular in clinical settings. Lajoie (2004) reported postural training could significantly improve the postural control in elderly subjects in combination with a feedback fading protocol [28]. Wolf et al. (1997) also analyzed the effectiveness of computerized balance training and revealed greater stability after training in older participants [29]. Patrice (2009) provided the participant's COP from a force platform through the screen of a monitor and found that the visual feedback from COP enabled participants to optimize their balance performance [30].

In these preceding studies, we compared the COP and the head displacement according to the presence of augmented reality to verify the effect of providing visual feedback in real time through AR HMD. Anterior-posterior head displacement in the single leg stance was significantly smaller than the displacement without feedback.

We estimate the reasons for these results as follows. Firstly, the size of the sample is too small to be confident in the results of statistical tests. It might have limited the power of our study to uncover the differences between the subjects. Secondly, the level of balance ability of each participant varied. Pre-inspection, according to the balance ability, could prevent the effects of differences among samples. Thirdly, the participants in this experiment were all healthy adult males. Therefore, we assume that augmented reality feedback may not have had significant effects on postural control during the test on a stable surface. We expect that the effects of augmented reality feedback will be more dramatic when performing postural tasks on an unstable surface (foam pad). And people with balance dysfunction exhibit greater postural sway under the same experimental conditions and the effectiveness of AR feedback will also be more pronounced. Lastly, discomfort caused by wearing an AR HMD might affect postural perturbation. Apart from participants' balance ability, performing a postural task with 579 g of AR HMD on could cause postural instability. In future studies, a balance performance test should be added to the criteria for the selection of participants, and experiments should be conducted after 30 or more participants are secured. Despite the results, we confirmed the applicability and potential of the AR HMD-based balance rehabilitation method we proposed.

The participants in this study that expressed interest in AR feedback during testing reported that it was easier to maintain balance through AR feedback, and some reported that moving an augmented object to a reference position was more difficult because they felt it was another task.

The clinical assessment of postural instability is important to establish a reliable differential diagnosis and develop optimal treatment strategies for patients with balance impairment. The standard in the clinical assessment of balance is a combination of history taking and clinically used balance tests, but neither approach is exhaustive as patients often forget or deny their history, and clinical tests are hampered by their subjective scoring and variable execution. In contrast, posturography can

quantify and objectively analyze postural responses. Increased body sway is a widely used indicator of postural instability. Unfortunately, a computerized posturography system is not feasible for most clinical or home setting. Because of expensive costs and time-consuming processes for testing, and the system requires an expert to acquire and analyze data. In these regards, the AR HMD-based method could help the therapist or clinicians to assess postural stability quantitatively and provide a training environment for patients to immerse. Furthermore, if various rehabilitation contents for AR HMD are developed, the patient could be able to train balance by themselves at their home without difficulty.

Judging from these results, the AR HMD-based balance rehabilitation method proposed in this study will be a new alternative to improving the existing complex and boring rehabilitation environment.

## 6. Conclusions

Recently, intelligent IoT and big data systems have been increasing in the medical and rehabilitation domain [54], so research related to the AI-based smart services and systems is delivering significant contributions in the fields of healthcare domain applications [55]. In line with this trend, a modern concept for rehabilitation using augmented reality is needed that links sensing data and virtual content with IoT, Big Data, and smart services.

This paper described a method for balance rehabilitation using an AR HMD and implemented the method to verify the method. This method assists therapists to assess their patients quantitatively and to provide an amusing training environment. We implemented the proposed method using the HoloLens and verified the method through an experiment. We assumed that during the postural task, there would be a positive correlation between the path length of COP measured through the WBB and the path length of head displacement measured through the HoloLens. And we also assumed that the path length of COP and the path length of head displacement when augmented reality feedback was provided would be smaller when compared to when augmented reality feedback was not provided. We have engaged young people with experience in using a HoloLens in this study before studying the elderly. Although the statistically significant results were few, we identified the potential of AR HMD as a reliable balance rehabilitation tool.

To validate the proposed method as an assessment tool that can be applied in the actual clinical environment, further verification experiment aimed at specific groups of patients (such as the elderly, patients with neurological disorders) is essential. To further extend this work, it will be necessary to apply the method to clinical friendly tests of balance (Romberg test, functional reach test, Berg Balance Scale) and to compare with the results from therapists and clinicians. In future studies, additional auditory feedback linked to head movements will be implemented, and the effectiveness will be verified by combining it with visual feedback.

During this work, we were faced with challenges such as the weight of the device and motion sickness, which affect the user's postural stability. To apply to clinical environments, hardware improvements must be accompanied.

We expect the proposed method will be employed as a convenient and effective rehabilitation tool for both patients and therapists in the future.

**Author Contributions:** E.-Y.L. conceived and designed the study. E.-Y.L. and V.T.T. analyzed the data. All authors contributed to data interpretation. All authors wrote the manuscript. D.K. critically revised the manuscript, and all authors approved the final version of the manuscript.

**Funding:** This work was supported by the MSIT (Ministry of Science and ICT), Korea, under the ITRC (Information Technology Research Center) support program [grant number IITP-2018-2018-0-01419].

**Acknowledgments:** This research was supported by the MSIT (Ministry of Science and ICT), Korea, under the ITRC (Information Technology Research Center) support program (IITP-2018-2018-0-01419) supervised by the IITP (Institute for Information & communications Technology Promotion).

**Conflicts of Interest:** The authors declare that they have no conflict of interest in the research.

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
