# Peer review of "A Novel Head Mounted Display Based Methodology for Balance Evaluation and Rehabilitation"

_sustainability, doi:10.3390/su11226453_

Round 1
Reviewer 1 Report
I would suggest not to include acromysms in the title
Abstract.
Data should be included in the abstract
Introduction
Include a reference that sustain your argument “Balance dysfunctions are general reasons for falls in elderly.” (P1, L42) and also for “Balance-impaired people tend to rely on vision and exteroceptive information on postural control.” (P2,L62)
The rationale behind the use of HMD combined with AR should be expanded.
Methods
Please include references supporting the reliability of the used devices (P3,L93)
Manufacturer, city, etc., must be reported for each device.
About Data Collection, it is necessary that you include the support when you use the analysis of the data.
Please describe the characteristics of the sensors integrated in the Hololens IMU
The use of Wii Balance Board to collect postural sway must be justified.
How both systems were integrated?
If the introduction is centred on the need of this technology in older adults, why don´t you validate it in this population group?
In the procedure please indicate the number of attemps, rest between attemps, outcome meassures…
How you described the normality of the data and please justify the use of Pearson correlations moments.
How the positions in the platform were determined?
Results
Please justify the large CIs
Please justify why only significant results were observed in Double leg stance
In my opinion to base the results section in correlations obtained from only 8 participants with different characteristics can be a limitation. Please comment.
If non-significant differences were observed when results with or without AR are compared, how you justify the need of this technology?
Discussion
I would suggest discussing the clinical and practice implications of the study findings more?
The discussion should be based on the Author´own results and not merely on other studies.
The conclusions are diffuse
References should be updated.
Author Response
Point 1: I would suggest not to include acromysms in the title 

Response 1: “HMD” has been amended to “Head Mounted Display”
Point 2: Data should be included in the abstract
Response 2: Abstract has now been amended.
Point 3: Include a reference that sustain your argument “Balance dysfunctions are general reasons for falls in elderly.” (P1, L42) and also for “Balance-impaired people tend to rely on vision and exteroceptive information on postural control.” (P2,L62)
Response 3: Citations have now been updated.
Point 4: The rationale behind the use of HMD combined with AR should be expanded.
Response 4: The purpose of this paper is to propose a rehabilitation method that utilizes AR HMD, which can evaluate and train balance at the same time with single device. We maintain that these attempts are important because AR HMD is expected to become lighter and cheaper in the future.
Point 5: Please include references supporting the reliability of the used devices (P3,L93).
Response 5: Citations have now been updated.
Point 6: Manufacturer, city, etc., must be reported for each device.

Response 5: Please see page 5, line 146 (3.1 Experimental Setup).
Point 7: About Data Collection, it is necessary that you include the support when you use the analysis of the data.
Response 7: We used test stances for experiment from the Balance Error Scoring System, a frequently used balance test in clinical setting, to verify the proposed method. Relevant content was added to 3.5. Procedure. Please see page 7, line 218.
Point 8: Please describe the characteristics of the sensors integrated in the Hololens IMU.
Response 8: Microsoft does not disclose the detailed specifications of IMU configuration sensors, and we have added a description of head tracking with IMU. Please see page 4, line 124.
Point 9: How both systems were integrated?
Response 9: The head displacement recording and augmented reality feedback in HoloLens are integrated in single software for HoloLens OS. The data from Wii Balance Board were recorded simultaneously with HoloLens through another developed software and then integrated with HoloLens data. Please see page 5 (3.2. Software Algorithm).
Point 10: If the introduction is centred on the need of this technology in older adults, why don´t you validate it in this population group?

Response 10: The authors would like to thank the reviewer for this suggestion. In this experiment, it was very difficult to recruit senior or patients for experiments where safety was not verified. We will recruit the targets again after completing the verification of safety.
Point 11: In the procedure please indicate the number of attemps, rest between attemps, outcome measures.
Response 11: The participants performed each test condition three times in a randomized order. Each trial lasted 20 seconds, and it began when the test position was held stable for several seconds. Between each test, participants were asked to sit down and take enough rest. You can find detailed information in page 7, line 234 (3.5 Procedure).
Point 12: How you described the normality of the data and please justify the use of Pearson correlations moments.
Response 12: We used Shapiro–Wilks test for normality and we have now added about this (page 8, line 242).
Nowadays, ICC has been widely used in conservative care medicine to evaluate interrater, test-retest, and intrarater reliability. To evaluate the interrater among devices, we refer to Koo, T. K., & Li, M. Y. (2016).
Koo, Terry K., and Mae Y. Li. "A guideline of selecting and reporting intraclass correlation coefficients for reliability research." Journal of chiropractic medicine 15.2 (2016)
Following this guideline, we used the formula for ICC (3, 1).
Point 13: How the positions in the platform were determined?
Response 13: There were no physical restrictions on the location of the installation, as both HoloLens and Wii Balance Board are wireless device connected to the computer via Bluetooth or Wi-Fi. Wii Balance board was installed on a hard, flat surface to measure the ground reaction force.
Point 14: Please justify the large CIs

Response 14: Firstly, the size of the sample is too small to be confident on the results of statistical tests. Secondly, the level of balance ability of each participant varied.
In future studies, balance performance test will be added to the criteria for selection of participants, and experiments will be conducted after 30 or more participants are secured.
We have now added about this to Discussion (page 11, line 321 ).
Point 15: Please justify why only significant results were observed in Double leg stance.
Response 15: The double leg stance condition is a "stable" condition that reduces postural sway. Therefore, we assume that stable test condition made low postural sway values on both HoloLens and Balance Board and resulted in significant positive correlations.
We have now added about this to Discussion (page 11, line 306).
Point 16: In my opinion to base the results section in correlations obtained from only 8 participants with different characteristics can be a limitation. Please comment.
Response 16: The authors agree with reviewer's point. As we written in point 14, the level of balance ability of each participant varied, and we will add pre-balance test to criteria for participant selection.
Point 17: If non-significant differences were observed when results with or without AR are compared, how you justify the need of this technology?
Response 17: The participants in this experiment were all healthy adult males. Therefore, we assume that augmented reality feedback may not have had a significant effectiveness on postural control during the test on stable surface.
We expect that the effects of augmented reality feedback will be more dramatic when participants with balance dysfunction or when performing postural task on unstable surface (foam pad).
Point 18: I would suggest discussing the clinical and practice implications of the study findings more?
Response 18: The authors agree with reviewer's point, and we added to the reason why clinicians need tool, which is portable, easy to install, also allow quantitative assessment and training for balance at the same time. Please see page 12, line 339.
Point 19: The discussion should be based on the Author´own results and not merely on other studies.
Response 19: According to the above reviewer’s points, the authors added interpretations of the results.
Point 20: The conclusions are diffuse
Response 20: This section has now been amended to be more exact.
Point 21: References should be updated.
Response 21: Reference has now been added.
Reviewer 2 Report
This paper presents a method for using visual feedback for balance rehabilitation purposes. However, the research question/s are not clearly defined and is it therefore hard to understand what is the contribution to science the authors would like to publish. The manuscript also needs proof-reading. Some further comments can be found bellow.
Abstract.
Line 9: Consider rewriting this sentence, check grammar; Maybe you could simply write: In this paper we present a new augmented reality…
Line 12: What does the visual feedback train?
Line 13: Which applications? What do they do?
Line 15: “Repeatedly« or »consecutively”?
Line 15: “whether augmented reality exists”? Did you maybe mean presence or absence of augmented reality?
Line 19: In the previous sentence the authors state most results were not significant, mainly due to the very small sample of test users. I am not sure this can than conclude that the authors have confirmed the applicability of the proposed platform if the results from the study suggest otherwise.
Line 20: Consider changing “proposed method will be” to “proposed method could be”
Line 20: In the first sentence you state that this paper presents a new “platform”, whereas you conclude it with “proposed method”. What is the paper presenting, the platform or the method how to use it?
Introduction
Line 26: Fall and fall related injuries can be fatal, not all of the are fatal for people aged 65 and older.
Line 28-29: Fear of falling and loss of confidence are rather a consequence of falling, and are not actually included in the fall.
Line 32: Check grammar, consider rewriting.
Line 52: Did you maybe think: “Using biofeedback training has shown to be a promising method to …”
Methodology
Line 93: In the abstract the authors use the terms “platform” and “method”, in this chapter they use “methodology” and “system”. I suggest choosing one term (the one that the authors believe best describes their contribution to science), clearly define it and use is consistently.
Line 204: In the introduction, the authors state that falls are mostly critical for the elderly and that their solution could be used to prevent such falls. However, the test group is of average age of only 24.75 years old. Is this a good representative test sample? Do these age groups have the same perception of new technologies such as AR?
Could you justify the very small sample of only 8 participants?
The tasks should be put in a separate sub-chapter Tasks. Could you please describe them in more detail? How was the task completion time defined? How did you define the task completion success?
The authors should also clearly state the dependent variables they observed in the study and in which unit were they measured/how were they observed.
The research questions should be also stated
Results and Discussion
Although the results do not show any significant difference between the success rate of completing the tasks when using AR or not, the authors discuss this study has confirmed the the applicability of the method with visual feedback.
Author Response
Point 1: Line 9: Consider rewriting this sentence, check grammar; Maybe you could simply write: In this paper we present a new augmented reality…
Response 1: This has now been corrected as suggested.
Point 2: Line 12: What does the visual feedback train? / Line 13: Which applications? What do they do?
Response 2: Visual feedback is largely used in rehabilitation to improve the control of posture and to train the ability to shift weight by moving the entire body or the trunk. Many devices used in the clinical practice provide training based on the feedback of a cursor on a computer screen, controlled by the position of the Center of Pressure (CoP) or of the Center of Mass (CoM).
Point 3: Line 15: “Repeatedly« or »consecutively”? / Line 15: “whether augmented reality exists”? Did you maybe mean presence or absence of augmented reality?
Response 3: This has now been amended and corrected.
Point 4: Line 20: Consider changing “proposed method will be” to “proposed method could be”
Response 4: This has now been corrected as suggested.
Point 5: Line 20: In the first sentence you state that this paper presents a new “platform”, whereas you conclude it with “proposed method”. What is the paper presenting, the platform or the method how to use it?
Response 5: This has now been unified to "method".
Point 6: Line 26: Fall and fall related injuries can be fatal, not all of the are fatal for people aged 65 and older.
Response 6: This has now been amended.
Point 7: Line 28-29: Fear of falling and loss of confidence are rather a consequence of falling, and are not actually included in the fall.
Response 7: This has now been amended.
Point 8: Line 32: Check grammar, consider rewriting.
Response 8: This has now been amended.
Point 9: Line 52: Did you maybe think: “Using biofeedback training has shown to be a promising method to …”
Response 9: This has now been corrected as suggested.
Point 10: Line 93: In the abstract the authors use the terms “platform” and “method”, in this chapter they use “methodology” and “system”. I suggest choosing one term (the one that the authors believe best describes their contribution to science), clearly define it and use is consistently.
Response 10: This has now been unified to "method" as suggested.
Point 11: Line 204: In the introduction, the authors state that falls are mostly critical for the elderly and that their solution could be used to prevent such falls. However, the test group is of average age of only 24.75 years old. Is this a good representative test sample? Do these age groups have the same perception of new technologies such as AR?
Response 11: The authors would like to thank the reviewer for this suggestion. In this experiment, it was very difficult to recruit elderly or patients for experiments where safety was not verified. We will recruit the targets again after completing the verification of safety. In addition, enough consideration is also needed for user experience design for augmented reality users who are elderly or patients.
Point 12: Could you justify the very small sample of only 8 participants?
Response 12: We needed participants with experience in dealing with augmented reality HMD for a smooth experiment. And it was practically difficult to recruit participants who had used augmented reality HMD. Future research will focus on verifying the proposed method and increase the number and age range of participants.
Point 13: The tasks should be put in a separate sub-chapter Tasks. Could you please describe them in more detail? How was the task completion time defined? How did you define the task completion success?
Response 13: This has now been amended. We asked the participants to maintain the test posture for 20 seconds per measurement. The task's failure was breaking the posture and falling but there was no one who fell. Please see page 7, line 225.
Point 14: The authors should also clearly state the dependent variables they observed in the study and in which unit were they measured/how were they observed.
Response 14: The dependent variables of this study are the path length of COP and the path length of head displacement. The unit of the variables are centimeter. The WBB and HoloLens measure the coordinates of the COP and the coordinates of the head position for each frame and we converted them into centimeters. For more detail, please see page 5(3.2. Software Algorithms).
Point 15: The research questions should be also stated
Response 15: We assumed that during the postural task, there would be a positive correlation between the path length of COP measured through the WBB and the path length of head displacement measured through the HoloLens. And we also assumed that the path length of COP and the path length of head displacement when augmented reality feedback was provided would smaller when compared to when augmented reality feedback was not provided. We have now added research questions (page 12, line 364).
Point 16: Although the results do not show any significant difference between the success rate of completing the tasks when using AR or not, the authors discuss this study has confirmed the the applicability of the method with visual feedback.
Response 16: Thank you for your kind words about our paper. We are delighted to hear that you think our work is meaningful in our field.
Round 2
Reviewer 1 Report
The reviewers overall are satisfied with the manuscript. While there are some issues with the signal processing and data analysis, they believe that it can be accepted in the present form.
Author Response
Thank you for your kind words about our paper. We are delighted to hear that you satisfied with the manuscript.

Reviewer 2 Report
I think the authors have tried to answer to all of my comments. While I am happy they decided to accept most of them, I still have major concerns about two things:
1) The authors claim, that with this research they confirmed the applicability of the proposed method although there are very few statistically significant results to support this claim. The explanation that most results were not significant, mainly due to the very small sample of test users just raises an additional question about the number of participants simply not being enough to publish this results.
2) (related also to comment no.1) The number and age of participants is too small/not appropriate for this study. Since you are proposing a new method, which is intended for the elderly, I still believe it should be tested with older participants, which would also provide more reliable user acceptance and user experience ratings.
Author Response
I agree to some extent with reviewer's comment.
However, we hope you understand that it was very difficult to find participants who had previously used augmented reality HMD.
We have engaged people with experience in using a HoloLens in the study.
This is because it takes a lot of time for users who first encounter a HoloLens to wear it familiarly and clearly see the augmented image on the display.
Therefore, it is almost impossible to find elderly people with experience using a HoloLens.
The study also aimed to apply and evaluate rehabilitation methods implemented for young adults before studying the elderly.
We thank the reviewers for their comments.
